# Role of Extracellular Matrix in Gastrointestinal Cancer-Associated Angiogenesis

**DOI:** 10.3390/ijms21103686

**Published:** 2020-05-23

**Authors:** Eva Andreuzzi, Alessandra Capuano, Evelina Poletto, Eliana Pivetta, Albina Fejza, Andrea Favero, Roberto Doliana, Renato Cannizzaro, Paola Spessotto, Maurizio Mongiat

**Affiliations:** 1Department of Research and Diagnosis, Division of Molecular Oncology, Centro di Riferimento Oncologico di Aviano (CRO) IRCCS, 33081 Aviano, Italy; eandreuzzi@cro.it (E.A.); acapuano@cro.it (A.C.); evelina.poletto@cro.it (E.P.); epivetta@cro.it (E.P.); albinafejza@gmail.com (A.F.); favero.andrea.7@gmail.com (A.F.); rdoliana@cro.it (R.D.); pspessotto@cro.it (P.S.); 2Department of Clinical Oncology, Experimental Gastrointestinal Oncology, Centro di Riferimento Oncologico di Aviano (CRO) IRCCS, 33081 Aviano, Italy; rcannizzaro@cro.it

**Keywords:** extracellular matrix, endothelial cells, angiogenesis, tumor microenvironment

## Abstract

Gastrointestinal tumors are responsible for more cancer-related fatalities than any other type of tumors, and colorectal and gastric malignancies account for a large part of these diseases. Thus, there is an urgent need to develop new therapeutic approaches to improve the patients’ outcome and the tumor microenvironment is a promising arena for the development of such treatments. In fact, the nature of the microenvironment in the different gastrointestinal tracts may significantly influence not only tumor development but also the therapy response. In particular, an important microenvironmental component and a potential therapeutic target is the vasculature. In this context, the extracellular matrix is a key component exerting an active effect in all the hallmarks of cancer, including angiogenesis. Here, we summarized the current knowledge on the role of extracellular matrix in affecting endothelial cell function and intratumoral vascularization in the context of colorectal and gastric cancer. The extracellular matrix acts both directly on endothelial cells and indirectly through its remodeling and the consequent release of growth factors. We envision that a deeper understanding of the role of extracellular matrix and of its remodeling during cancer progression is of chief importance for the development of new, more efficacious, targeted therapies.

## 1. Introduction

Gastrointestinal (GI) cancers represent one of the major causes of cancer-related deaths worldwide. This group of tumors comprises colon, stomach, and liver cancers which altogether account for most of the fatalities [1]. In addition, despite being less common, the group includes also the pancreatic ductal adenocarcinoma, characterized by an extremely poor prognosis, and esophagus cancer [1,2]. Like all cancers, GI tumors arise following the acquisition of oncogenic mutations and, for the stomach, small intestine, and colorectum, the likelihood of this event could be linked to the short lifespan of epithelial cells, which must be replaced every few days to overcome the physical, chemical, and biological insults [3]. GI patients are often diagnosed at advanced stages of the disease, which dramatically reduces the chances of effective cures. Thus, to improve the survival rate of these patients there is a need to develop new therapeutic strategies.

In the past years, an extensive effort has been drawn to characterize the driving mutations of tumor cells in order to employ more specific drugs, with only partial positive responses in a small percentage of gastric cancer (GC) patients [4,5]. Despite vast preventive screening that has improved the five-year survival rate of colorectal cancer (CRC), metastatic CRC still represents a major health threat. Targeted therapies, such as the use of epidermal growth factor receptor (EGFR) inhibitors, provide only partial responses and for a limited period of time [6,7], with the Kirsten-rat sarcoma 2 viral oncogene homolog (KRAS) mutation status being predictive of the response [8]. In addition, a peculiar trait of CRC is that its natural history is highly dependent on the inflammatory processes that are often at work in this tumor.

An increasing number of studies have demonstrated that the nature and composition of the tumor microenvironment (TME) can profoundly affect tumor progression and represents an important mean for the development of new therapeutic approaches to overcome this clinical stumble [9,10]. In fact, tumor progression is the result of a continuous crosstalk between cancer cells and the surrounding microenvironment. The TME is composed of a variety of cells including endothelial cells (ECs), fibroblast, and immune cells, and by secreted proteic components such as cytokines, growth factors, and the extracellular matrix (ECM), with the normal as well as the neoplastic cellular components being nourished by the newly formed vascular network [11]. These components exchange a large amount of molecular information with cancer cells that not only affect their growth and progression but also dramatically influence the therapeutic response and the patient’s outcome [12]. Despite the inflammatory response being generally meant to destroy neoplastic cells, once recruited inflammatory cells can be reprogrammed by neoplastic cells and by the TME into a tumor-promoting phenotype [13,14]. Cancer-associated fibroblasts and macrophages significantly affect the development of GI tumors [15,16] and immunotherapy has been shown to be effective, at least in a subset of CRC patients [17,18,19].

Given the above scenario, a more accurate understanding of the relationships of the crosstalk between cancer cells and the TME cellular and stromal components may grant the possibility to improve the prognostic accuracy and treatment choices for GI cancer patients.

As previously mentioned, an important component of the TME is the ECM. The importance of the ECM is highlighted by the fact that it can influence all the hallmarks of cancer identified by Hanahan and Weinberg [20], impinging on restless proliferation, evasion from tumor suppression, replicative immortality, resistance to cell death, initiation of cell invasion, abnormal cellular energetics, and avoidance of the immune response and chronic inflammation, as well as induction of angiogenesis [21]. Once seen as a mere scaffold for cells, providing mechanical and structural support, the ECM can instead profoundly influence cell function and its dysregulation during cancer growth deeply affects cancer progression [22,23,24]. For example, the degradation of the basement membrane (BM) components of ECM represents a crucial step during GI tumor development [25,26].

Of particular interest is the purpose of ECM in tumor angiogenesis, a pathological process engaged by tumor cells to overcome the shortage of oxygen and nutrients [27,28,29]. In the last few decades, extensive efforts were undertaken to develop new tools to target ECs, with the advantage that these cells are more stable compared to cancer cells. These endeavors have led to the discovery of several anti-angiogenic proteins, some of which were brought to clinical trials. These angiostatic factors are often the result of proteolytic degradation of ECM constituents and act both directly on ECs, through the binding to receptors on the EC surface, or indirectly modulating the growth factor availability [30]. Given its strategic role the remodeling of the ECM is finely regulated [24,31]. In this survey, we will review the current knowledge on how the ECM regulates angiogenesis in the context of GC and CRC. These tumor types respond differently to anti-angiogenic therapy and the use of the vascular endothelial growth factor A (VEGFA) blocking agent bevacizumab is currently applied only in metastatic CRC [32,33,34], further highlighting the key role of the TME in the regulation of tumor angiogenesis and the therapy response.

## 2. ECM and Angiogenesis: Highlights in GI Cancers

The ECM plays a renowned function in angiogenesis and several are the molecules affecting EC in a positive or negative way. Thus, the effect that ECM exerts in the angiogenic process is complex and is the result of the balance between promoting and hindering forces, depending on the relative concentration of the molecules in a given tissue. Here, we will deal with the ECM molecules which are most relevant in the vascularization of GC and CRC, and a schematic representation of the major ECM molecules affecting GI tumor-associated angiogenesis is reported in Figure 1.

Laminins (LMs) are major components of the epithelial and endothelial basement membrane (BM) and play an important role in the regulation of tissue homeostasis [35,36,37] and vessel stability [38]. Several LM isoforms are expressed in the human intestine, including LM-111 (the numbers refer to the α, β, and γ chain isoform, respectively), LM-511, and LM-332 with specific deposition patterns along the crypt-villus axis during development and in the blood vessels [39,40]. As further evidence of the importance of LMs in this tissue, the expression of the LM α1 and α5 chains (LMα1 and LMα5, respectively) in patients affected by intestinal bowel disease (IBD) are upregulated, suggesting that they may play an important role in the microenvironmental response to inflammation [41]. Of note, LMα1 is over-expressed in CRC [42] and associates with increased recruitment of stromal cells, angiogenesis, and enhanced CRC growth. LMα1 engages an intimate crosstalk between cancer, stromal, and endothelial cells through the recruitment of cancer-associated fibroblasts (CAFs) and the induction of VEGFA production via the integrin α2β1-CXCR4 (C-X-C chemokine receptor type 4) complex. A recent study demonstrated that increased the laminin subunit α-5 (LAMA5) gene expression in tumor cells upregulates Notch signaling, affects EC branching, and inhibits metastatic CRC growth [43]. These results demonstrate that cancer cells start to produce specific LM chains as a strategy to promote angiogenesis and to facilitate metastatic spread; thus, the inhibition of vascular BM laminins’ expression may represent a useful therapeutic approach to impair the development of hepatic metastases of CRC.

Collagens, the main structural proteins of the ECM, are known to profoundly affect EC survival and vessel formation and it is well known that angiogenesis depends on proper collagens’ biosynthesis and cross-linking [44,45]. The major players of this family are the interstitial type I and the BM-associated collagens type IV, XV, and XVIII. However, a striking imbalance of ECM composition is observed in GI malignancies and, particularly, in CRC where an increased expression of type I collagen is accompanied with the reduction of type IV [46,47]. It has been suggested that these modifications may impact on EC behavior and tumor angiogenesis.

The BM collagens-derived matrikines, bioactive fragments originated through degradation, have been broadly studied in the context of GI disorders where extensive turnover of BM takes place. In fact, an increased release of type IV collagen fragments can be detected in the blood stream of CRC patients [48], as well as in other GI disorders [49], that may function as putative prognostic biomarkers of the diseases [50].

The best described matrikine is endostatin, the noncollagenous domain 1 (NC-1) domain of type XVIII collagen α1 chain, which has been extensively characterized for its anti-angiogenic and anti-tumor activities [51]. In a wide number of clinical studies recombinant human endostatin (Endostar) was successfully employed in the treatment of CRC and GC [52,53]. Liang and colleagues employed a *Salmonella typhimurium* mutant (S636) to specifically deliver endostatin in colon cancer cells, proposing a new intriguing therapeutic strategy [54], and the use of endostatin was proven effective also in patients with liver metastasis [55].

Among the type IV collagen-derived matrikines, arresten, an antiangiogenic bioactive fragment derived from the α1 chain of collagen IV, reduced CRC growth and decreased microvessel density in preclinical settings [56], but it has not been assessed as possible therapeutic agent in human GI malignancies yet.

Canstatin, a type IV collagen α2 chain-derived matrikine able to reduce tumor growth and blood vessel density, has been successfully employed in the treatment of CRC [57] and GC [58] xenograft models.

Similar properties are shared by tumstatin, another type IV collagen fragment [59,60]. However, the over-expression of endogenous inhibitors like tumstatin, but also endostatin or thrombospondin-1 (TSP-1), induces the upregulation of angiopoietin-2, basic fibroblast growth factor (bFGF) and platelet-derived growth factor type A (PDGF-A) in colon cancer cells as a mechanism of escape from anti-angiogenic stimuli [61], and the use of these fragments is more effective when used in combination with other angiogenic inhibitors or conventional chemotherapy/radiotherapy. Wei and colleagues cleverly exploited the *Bifidobacterium longum*, which localizes and proliferates in the hypoxic angiogenic environment surrounding solid tumors for tumstatin-targeted delivery, showing significant reduction of tumor growth and microvessel density [62].

Another ubiquitous ECM component is fibronectin, a multifunctional ECM glycoprotein with established pro-angiogenic function [63,64,65,66,67,68,69], which is often upregulated in several solid cancers, including CRC [70]. Its alternatively spliced extra domain A (EDA), considered a marker of angiogenesis and tissue remodeling, is of particular interest in the GI context being over-expressed in dextran sodium sulfate (DSS)-induced experimental colitis models and in patients with inflammatory bowel disease (IBD) [71]. CRC cell-derived EDA plays a key role in promoting tumor vasculogenesis and metastatic spread [72,73,74,75]. Interestingly, EC-derived fibronectin expressing the EDA domain enhances the metastatic capacity of CRC cells inducing the epithelial–mesenchymal transition, highlighting the role of ECM in modulating a pro-active role of EC on tumor cell behavior [76].

By virtue of their polyhedric nature, proteoglycans affect the development of a wide variety of tumors, including GI malignancies. However, only a few studies demonstrate a direct implication of these molecules in GI cancer-associated angiogenesis.

The proteoglycan perlecan and its carboxy-terminal fragment endorepellin best describe the fundamental balance required to maintain a functional angiogenic system. Unprocessed perlecan exerts a pro-angiogenic function, whereas its carboxy-terminal fragment endorepellin halts angiogenesis [77,78,79,80]. Perlecan prompts tumor growth and angiogenesis in CRC [81], and its interferon (IFN)-γ-mediated transcriptional repression may halt the induced angiogenic stimulus [82]. A more recent study included perlecan among the ECM molecules that are downregulated by sulindac treatment in CRC cells [83]. Endorepellin can further be processed by the bone morphogenetic protein 1 (BMP1)/Tolloid-like protease and releases the bioactive angiostatic laminin G-like domain 3 (LG3) domain, which can also be found in the secretome of CRC cells suggesting that it may play an important role in this context [84].

Biglycan is another proteoglycan whose expression is altered in CRC, among other cancers [85]; a high expression of this small leucine-rich proteoglycan correlates with poor prognosis [86]. The pro-angiogenic effects of biglycan are exerted through the extracellular signal-regulated kinase (ERK)-mediated release of VEGFA [87], thus it may represent an important target for anti-angiogenic therapy.

Hyaluronan, also referred to as hyaluronic acid (HA) or hyaluronate, is a naturally occurring nonsulfated glycosaminoglycan (GAG) component of connective, epithelial, and neural tissues. It plays a pivotal role in cancer and its deposition is altered in experimentally induced colitis [88,89]. Hyaluronan synthesis is finely regulated in vascular cells, highlighting its important role in angiogenesis [90]. Elevated levels of the hyaluronan synthase 1 gene correlate with poor prognosis of CRC patients [91,92], and its blockage may represent an encouraging therapeutic approach [93].

The role of thrombospondin-1 and -2 (TSP-1 and TSP-2, respectively) as potent endogenous angiogenesis inhibitors has been extensively studied. Interestingly, TSP-1 expression is reduced in CRC and inversely correlates with tumor vascularity and prognosis [94,95,96]. Of note, high levels of VEGFA combined with the absence of TSP-1 associate to worse outcome in these patients [94,95]. Jo and colleagues demonstrated a gradual wingless/integrated (Wnt)-dependent decrease of TSP-1 expression with tumor progression being almost undetectable in invasive adenocarcinomas [97]. In addition, TSP-1 loss in GI malignancies has also been ascribed to promoter hypermethylation [98,99] or messenger ribonucleic acid (mRNA) depletion through the microRNAs miR-18, miR-19, and miR-194 [100]. Intriguingly, TSP-1 null mice are more susceptible to DSS-induced colitis and display a robust inflammatory-driven angiogenic response [101,102], partially rescued by TSP-mimetic peptides [101,103]. These studies provide further proof to the current vision about the tight intertwining interaction between the inflammatory and angiogenic processes [104] and highlight the importance of investigating negative regulators of angiogenesis also in IBD for potential innovative therapeutic strategies. TSP-1 and TSP-2 also correlate with interleukin (IL)-10 expression, a further indication of the crosstalk between angiogenesis and inflammation; accordingly, CRCs expressing IL-10 are characterized by low vascular density suggesting that it may stimulate the expression of angiostatic factors [105].

Despite being less studied, compared to the aforementioned ECM molecules, the mindin/F-spondin family, and in particular spondin-2, displays a strategic function in GI tumor-associated angiogenesis. Through its pro-migratory effects, it promotes CRC metastatic dissemination [106] and regulates vessel formation affecting vascular smooth muscle cell proliferation and migration [107]. Spondin-2 expression is regulated by early growth response factor-1 (Egr-1) and it is decreased in CRC tumors [108]. Spondin-2 inhibits EC migration and tube formation and its over-expression impairs angiogenesis and tumor growth, suggesting that it may function as a potential target for anti-angiogenic therapy as well as a biomarker for CRC [108]. Intriguingly, also spondin-2 may be involved in the crosstalk between angiogenesis and inflammation since its expression is induced upon DSS-induced intestinal inflammation and in turn it activates the nuclear factor kappa-light-chain-enhancer of activated B cells (NF-κB) promoter through the Toll-like receptor 9 pathway [109]. Spondin-2 is upregulated in GC and it may regulate angiogenesis also in this tumor context, representing a possible biomarker and/or new therapeutic target for GC [110].

Multimerin-2 and Elastin microfibril interfacer 2 (EMILIN-2) are two ECM proteins, both belonging to the EMI domain endowed (EDEN) family [111,112,113,114], that display key functions in angiogenesis and have been extensively studied in the context of GI tumors. Despite their molecular affinity, multimerin-2 is an angiostatic molecule and functions as a gatekeeper of vascular stability, whereas EMILIN-2 promotes angiogenesis. Multimerin-2 is deposited specifically in tight association with the endothelium [115]. Importantly, multimerin-2 halts the activation of vascular endothelial growth factor receptor 2 VEGFR2 through the sequestration of VEGFA [116,117] and, thus, its expression may significantly affect the efficacy of anti-angiogenic therapy. Multimerin-2 is highly expressed along the blood vessels in the normal gastric and colonic mucosa whereas in many GI tumor-associated vessels the expression of the molecule is significantly altered [118,119]. The loss of multimerin-2 is partly due to the action of matrix metalloproteinase (MMP)-9 and MMP-2, the main MMPs activated during angiogenesis [120]. MMP-9-driven degradation of multimerin-2 occurred in CRC-associated vessels [118]. However, the loss of multimerin-2 may also be ascribed to a decreased expression of the molecule in response to high VEGFA levels during active angiogenesis [118,121], as detected in GC [122]. Importantly, multimerin-2 controls EC function and it is important for the maintenance of vascular stability. In fact, the downregulation of multimerin-2 in ECs causes the dismantlement of cell-cell junctions, leading to increased vessel permeability and leakage [123]. Thus, it is plausible to speculate that the loss of multimerin-2 could be associated with a worse drug delivery and poor prognosis for CRC and GC patients. The cognate protein EMILIN-2 exerts multiple functions in the TME overall, leading to tumor-suppressive effects [124,125,126,127]. Surprisingly, the over-expression of this matricellular protein in tumor xenografts led to the discovery of the pro-angiogenic function of EMILIN-2 [127]. This function is unfolded though the engagement of EGFR, and the consequent over-production of interleukin-8 (IL-8), which, in turn, stimulates EC proliferation and migration [128]. EMILIN-2 deposition nicely decorates the lamina propria of the gastric mucosa and its expression is significantly decreased in GC [119]. The EMILIN-2 effect in GC is dual, acting directly on GC cells to impair their proliferation and increasing the apoptotic rate and indirectly on EC and angiogenesis by enhancing the expression of angiogenic cytokines as serine protease inhibitor 1 (SERPINE-1), VEGFA, and IL-8 by GC cells [129]. Thus, being IL-8 at the crossroad between angiogenesis and inflammation and a crucial cytokine in GC-associated angiogenesis [130,131], EMILIN-2 may represent a keystone in the crosstalk between the two processes.

Like EMILIN-2, also tenascin-C (TNC) exerts multifaceted and angiomodulatory functions in the TME [132,133,134,135]. TNC was identified as a gene associated with IBD [136], and its serum levels in these patients correlate with the clinical and histological parameters of the disease [137]. TNC expression increases and contributes to the modulation of inflammation in DSS-induced acute colitis and in a mouse model of spontaneous Crohn’s disease [138], suggesting also for TNC a possible co-modulation of both angiogenesis and inflammation. TNC expression correlates with CRC malignancy [139] and may be considered a putative biomarker for metastasis [135,140]. Kawamura and colleagues demonstrated that TNC promotes colitis-associated cancer development through αvβ3-mediated angiogenesis and the disruption of this interaction is suggestive of promising therapeutic applications [141].

Another ECM molecule that modulates angiogenesis in the context of GI tumors is secreted protein acidic and rich in cysteine (SPARC)-like protein 1 (SPARCL1), a member of the SPARC family [142]. SPARCL1 is downregulated in various GI malignancies, such as CRC [143]. Its importance in angiogenesis is testified by the fact that it was firstly isolated from a human high endothelial venule (HEV) copy Deoxyribonucleic acid (cDNA) library [144] and it inhibited EC adhesion and spreading [145], resembling the anti-angiogenic activity exerted by its closest family member, SPARC/osteonectin [146]. Naschberger and colleagues, analyzing the transcriptome of tumor-associated EC, demonstrated that SPARCL1 expression is progressively lost in EC derived from CRC tumors, characterized by worse clinical prognosis [147]. This study demonstrated that SPARCL-1 regulates EC quiescence and vessels’ homeostasis by inhibiting proliferation, migration, and angiogenic sprouting, thus contributing to favorable prognosis.

Finally, also the role of netrins, laminin-like secreted proteins, should be mentioned, given their established role in angiogenesis and blood vessel network formation [148]. Netrins were originally identified as axonal guidance molecules and, in analogy with the nervous system, they act as bifunctional modulators in angiogenesis exerting both pro- and anti-angiogenic activities [149,150,151]. The majority of CRCs harbor defects in netrin-1 receptors, highlighting the importance of this regulatory pathway in the GI context. Netrin-1 induces angiogenic responses increasing nitric oxide production in ECs [152] and affects EC function and filipodia formation [150,153]. Netrin-4 was shown to promote angiogenesis in a model of post-ischemia revascularization [151] and may represent a potential alternative target to anti-VEGFA treatments [154]. In contrast with these findings, the over-expression of netrin-4 decreased angiogenesis and CRC growth [155], as well as lymph node and lung metastasis [156], further highlighting the bifunctional and context-dependent activity of these molecules.

Taken together, these studies highlight the prominent role of the ECM in GI tumor-associated angiogenesis and disclose the possibility to develop less toxic anti-angiogenic therapeutic approaches and/or predictive markers of therapy efficacy.

## 3. Impact of ECM on Endothelial Cells—Mechanisms of Action

The complexity of ECM is at the basis of the many different mechanisms through which it influences ECs’ biology. Here, we gathered the different mechanisms into three main groups: (1) Direct interaction of the ECM with EC surface receptors, (2) indirect effects due to the modulation of soluble growth factors, and (3) mechanisms activated by mechanical forces. A schematic representation of the main mechanisms by which the ECM affects EC function is reported in Figure 2.

### 3.1. Direct ECM-EC Interactions

ECs are highly dependent on the ECM-derived biochemical cues, which are mainly disclosed through the direct engagement of cell-surface receptors such as several integrins, VEGFR2, cluster of differentiation (CD)36, CD44, CD93, Hyaluronan-mediated motility receptor (RHAMM), and exert multiple and sometimes contrasting biological effects: EC proliferation and survival, adhesion, migration, chemotaxis, expression of hypoxia-inducible factor 1-α (HIF-1α) and/or VEGFA, and apoptosis.

Integrins, the main cell adhesion receptors, are crucial in mediating interactions between ECM components and EC, and a deep understanding of how these interactions affect angiogenesis may unveil the possibility to develop new therapeutic targets [157,158]. The major EC integrins engaging ECM ligands are α1β1, α2β1, αvβ3 and αvβ5, and their role was mostly investigated in the context of cancer [159]. The binding of collagen type I to the above mentioned integrins induces the activation of p44/p42 (Erk1/Erk2) mitogen-activated protein kinase (MAPK) pathway, resulting in EC proliferation and survival, and, thus, supporting tumor growth [160,161], whereas, collagen type IV promotes EC migration through integrins α1β1 and αvβ3 and activation of the focal adhesion kinase (FAK)-signaling cascade [162,163]. In contrast, its fragments, arresten, canstatin, and tumstatin, inhibit crucial pathways that lead to angiogenesis. Arresten prevents the binding of collagen type IV to the α1β1 integrin, resulting in MAP kinase pathway inhibition. In addition, it was also shown to inhibit FAK phosphorylation and HIF-1α and VEGFA expression [164,165]. Integrin engagement by canstatin leads to procaspase-9 activation through the inhibition of the FAK/PI3K (phosphoinositide 3-kinase) pathway [166]. On the other hand, tumstatin mediates the antiangiogenic activity via the interaction with integrin α3β1, leading to the inhibition of the hypoxia-induced cyclo-oxygenase 2 (COX-2) via the FAK/Akt (alpha serine/threonine-protein kinase)/NF-κB pathway [167,168,169].

Endorepellin [77] acts through the dual engagement of α2β1 and VEGFR2, causing rapid internalization and downregulation of both receptors, a mechanism referred to as dual receptor antagonism [170,171], ultimately leading to transcriptional repression of HIF-1α and VEGFA, and anti-tumor effects [172,173,174].

Also TSPs directly engage different integrins [175]. Binding of TSP-1 to α3β1 and α6β1 mediates adhesion and chemotaxis, whereas α4β1 ligation stimulates cell survival and proliferation [176]. However, the effects of TSP-1 on EC behavior have been shown to occur mainly through non-integrin receptors. In fact, both TSP-1 and TSP-2 contain thrombospondin type 1 repeat (TSR), which mediates the interaction with CD36 [177], resulting in the phosphorylation of cellular jun (c-jun) N-terminal kinases (JNK) and caspases and, therefore, EC apoptosis [178,179]. Increased caspases-3 activation and EC apoptosis is also induced by TNC, despite that the mechanism has not been elucidated [180,181], whereas netrin-1 halts EC apoptosis through the engagement of unc-5 netrin recptor B (UNC5B) and the blockage of the death-signaling effector death-associated protein (DAP) kinase [182,183]. Furthermore, through the engagement of CD47, but also CD36, TSP-1 inhibits the nitric oxide (NO)/cGMP (cyclic guanosine monophosphate)-signaling pathway [184,185]. The angiogenic role of HA and, in particular, the low molecular weight (LMW-HA) bioactive fragment, is exerted through a direct binding to CD44 and receptor for HA-mediated motility (RHAMM) [186], hence leading to mitogen-activated protein (MAP) kinase (ERK-1/2) activation and, consequently, EC proliferation [187].

Very recent investigations identified members of the C-type lectin domain-containing group 14 family as unique receptors for multimerin-2 [188,189]. In particular, CD93 is one of the most studied receptors of this family and plays an important role in the regulation of EC function [190]. The formation of the multimerin-2/CD93 complex is a prerequisite for the activation of integrin α5β1 and the consequent activation of the FAK cascade in ECs [191].

### 3.2. Indirect Mechanisms

Another important mechanism by which ECM proteins affect EC function is the interaction with growth factors and the consequent regulation of their distribution, availability, and presentation to the cognate receptors on the EC surface [192,193]. These multiple interactions are followed by a variety of responses (see Figure 2).

Among the growth factors that are secreted as precursors and need to be activated in the extracellular space, transforming growth factor β1 (TGF-β1) plays a key role in GI cancers [194]. TGF-β1 modulates EC proliferation, tube formation, and migration in a highly context-dependent manner [195], and is regulated at multiple levels, including secretion and interaction with ECM components [196]. Upon secretion as a homodimer, together with its latency-associated pro-peptide (LAP), it binds to the ECM and this interaction is further supported by covalent transglutaminase-induced crosslinks. The localization of latent TGF-β1 to the ECM is required for effective TGF-β1 activation. Even if TSP-1 is considered one of the major endogenous modulators of TGF-β1 activation [197], recently also TSP-4 has emerged as a key element in TGFβ1-evoked angiogenesis. Notably, TSP-4, whose expression is frequently increased in GC [198], exerts a pro-angiogenic function and associates with enhanced vascularization [199].

Once bound to the ECM, angiogenic cytokines are released depending on their binding affinity and the action of specific proteases [200]. Thus, the ECM releases signaling molecules at different kinetics and from different locations, allowing an extremely tight spatio-temporal regulation of cell fate within the TME [201]. The establishment of stable gradients through the binding with ECM molecules impacts on the activity of the key angiogenic factor VEGFA, which, depending on its concentration, can regulate various aspects of tumor angiogenesis [202,203]. VEGFA is upregulated in GI cancer and represents a useful prognostic and predictive biomarker [204,205,206]. In the TME, the majority of VEGFA is bound to ECM molecules as TSP-1 and -2 [175], perlecan [207], and multimerin-2 [116,117]. The degradation of these ECM proteins is necessary to allow the engagement of VEGFR2 and the cosequent autophosphorylation of the receptor leading to ERKs, p38 MAPK, and p125FAK activation [208]. This VEGFA-induced signaling produces several cellular responses in ECs including strong mitogenic and survival signals [209,210,211].

FGF-2 is cytokine that is tightly regulated through the binding with ECM and is altered in GI cancers [212,213]. The binding of FGF-2 to different ECM molecules results in peculiar biological effects: While the interaction of FGF-2 with fibronectin favors the engagement of its cognate receptors and the activation of the downstream pathway, [214], the binding to TSP-1 and -2 sequesters the cytokine, preventing its pro-angiogenic activity [215,216]. The absence of FGF-2 signaling results in the decoupling of vascular endothelial (VE-cadherin and p120-catenin, leading to the loss of adherens and tight junctions in ECs and increased vascular leakiness [217]. Once FGF-2 is released from ECM, it binds to fibroblast growth factor receptor (FGFR) on ECs and triggers the PI3K/Akt/mTOR (mammalian target of rapamycin) pathway, contributing to the maintenance of EC homeostasis [215].

The coexistence in the same ECM proteins of binding sites for many different growth factors concentrates such soluble mediators close to their own cell-surface receptors favoring receptor clustering and crosstalk. Due to its complex structure, the ECM molecule that mostly represents this function is perlecan, which can interact with many cytokines through its protein core or the heparan sulfate chains [207,218]. Beyond the already mentioned binding with VEGFA, perlecan interacts and modulates the activity of several growth factors including many of the FGF family (FGF-1, FGF-2, FGF-7, FGF-9, and FGF-18), FGF-binding protein, platelet-derived growth factor (PDGF), activin A, hepatocyte growth factor (HGF), and progranulin [219,220,221,222,223,224,225]. Collectively, perlecan exerts a pro-angiogenic function by presenting VEGFA and the various FGFs to their cognate receptors [79,217,226,227]. The simultaneous binding of perlecan to VEGFA and FGF-2 is of peculiar interest, given the crosstalk that occurs between these two signaling pathways [228].

The binding of growth factors to ECM molecules can also act by favoring or hampering the engagement of the specific tyrosine kinase receptor [229]. As an example, the binding of EMILIN2 with both EGFR and EGF promotes the activation of the downstream Janus kinase/signal transducer and activator of transcription (JAK/STAT3) pathway and leads to the overexpression of IL-8, a crucial cytokine regulating ECs’ proliferation and migration [127,128]. Also, in GC cells the activation of the EGF/EGFR pathway increases the production of angiogenic molecules [230,231], and this mechanism is influenced by EMILIN2 [119]. On the other end, endostatin exerts its anti-angiogenic function by hampering the binding of VEGFA to VEGFR2, thus resulting in decreased EC proliferation [232]. Moreover, the inhibition of VEGFA/VEGFR2 interaction by endostatin blocks VEGFA-induced EC migration by inducing endothelial nitric oxide synthase (eNOS) dephosphorylation [220].

Thus, the ECM is capable of integrating complex, multivalent signals in ECs in a spatially organized and regulated fashion. In this view, ECM-bound growth factors could be released locally or presented as complexes associated with the ECM. Such complexes could enhance membrane-proximal regulation among the receptors and influence the integration of the transduced signals.

### 3.3. Mechanical Cues

It is a well-established concept that the ECM is deregulated and disorganized in solid tumors and that the enhanced ECM stiffness is caused primarily by increased collagen deposition and enhanced crosslinking within the stroma and promotes cancer progression [233]. ECM remodeling results in heterogeneous three-dimensional matrix features, i.e., organization, rigidity, and composition. The changes of the ECM mechanical forces can significantly impact on EC signaling and behavior, resulting in the promotion of angiogenic processes [234]. In this view, the morphology and the dynamics of the sprouting vessels are controlled also by the chemo-mechanical and geometric properties at the capillary interface [235,236]. ECs sense and respond to mechanical cues through an interconnected system of mechanosensors that include integrins, cell-cell adhesion receptors, receptor tyrosine kinases (RTKs), and other membrane proteins such as ion channels and G-protein-coupled receptors. In the tumor-associated ECs, many of these molecules become deregulated, leading to altered cell functions. ECM stiffness induces integrin clustering in the focal adhesions, and the direct interactions between these integrins and RTKs leads to a spatial clustering of tyrosin kinase receptors [192]. This results in an alteration of the balance between ligand concentration and receptor auto-phosphorylation, which amplifies ligand-induced RTK signaling [237]. Additionally, the localization of RTKs within focal adhesions promotes the signaling through FAK, stimulating EC migration. Taken together, this suggests that extracellular cues from the surrounding ECM do not only amplify the signaling response from these receptors but also qualitatively change the functional outcome of RTK activation by altering the downstream transducer activation [192]. The specific mechanosensory pathways utilized by ECs to respond to aberrant mechanical cues are not yet well characterized, and much work still remains to fully understand the molecular mechanisms involved. Identifying these pathways will allow a better understanding of the mechanical regulation in tumor angiogenesis and provide new tools to govern the physical forces in tumors.

## 4. Function of ECM Remodeling in GI Tumor-Associated Angiogenesis

In the complex and dynamic process of angiogenesis, the breakdown and changes of the ECM are crucial events that reflect/impact on tumor cells but also on microenvironmental components such as ECs, impinging on their proliferation, migration, and differentiation and, consequently/ultimately, on the tumor vascularization. The major regulators of matrix turnover are secreted proteases, which consist of very large families of molecules. Their mechanism of action has been well studied in various type of malignancies, including CRC and, even if with a lower impact, GC.

MMPs are the most studied proteases involved in ECM degradation. By virtue of their role in cleaving the ECM, they facilitate EC migration through the matrix. In turn, ECM remodeling leads to the release of VEGFA or other ECM-bound angiogenic growth factors, or, conversely, the formation of collagens-derived protein fragments displaying anti-angiogenic activity. Thus, MMPs are recognized to play a key action in the context of angiogenesis (for a general review see [238]).

It is known that the local release of soluble VEGFA is a key event in angiogenesis. Following secretion, the bioavailability of the ECM-bound VEGFA is regulated through proteolytic cleavage aided by the acidic pH of the TME. In the colon microenvironment, MMP-2, MMP-7, MMP-9, and MMP-14 play a pivotal role being over-expressed or hyperactive [239]. However, a direct contribution to angiogenesis has been experimentally demonstrated only in few studies. Based on the observation that the levels of VEGFA and MMP-9 (but not MMP-2) correlated and were mutually increased, Hawinkels and colleagues suggested that MMP-9 could play a prominent response in the release and bioavailability of VEGFA in CRC. These authors showed that the neutrophils-derived MMP-9 within the CRC microenvironment mediates the release of biologically active VEGFA through the cleavage of heparan sulfate proteoglycans (HSPGs) [240]. In a cohort of 299 GC patients, the expression of MMP-2, MMP-9, and VEGFA positively correlated with the tumor size, invasive depth, lymphatic and venous invasion, lymph node metastasis and staging [241]. Despite that in this study there was no direct proof that the MMPs’ activity impinges on angiogenesis in GC, it was in accordance with the known prominent role of MMPs in the development of a vasculature permissive to tumor growth and metastatic spread.

Acting on different substrates, the outcome of MMPs is a balance between pro-angiogenic and anti-angiogenic effects; for instance, MMP-9 can also generate the angiogenic and tumor repressor tumstatin, a fragment including the noncollagenous domain (NC1) of collagen alpha-3 (IV) [242,243]. Once released, tumstatin reduces EC proliferation and induces apoptosis through the engagement of αVβ3 integrin [59]. On the other hand, CRC cells may be at least in part resistant to ECM-derived anti-angiogenic stimuli. In fact, Namali and colleagues demonstrated that tumors developed following the injection of colon cancer cells over-expressing tumstatin, endostatin, or TSP-1 finally escaped angiogenesis inhibition [61]. The combination of all three angiogenesis inhibitors displayed no additive effects compared to the over-expression of a single inhibitor, suggesting the presence of a functional redundancy in this system. Interestingly, this study demonstrates that parallel pro-angiogenic pathways can be engaged in CRC microenvironment, which may overcome the inhibitory pressure. This could also explain the mechanisms by which CRC patients often acquire resistance to anti-angiogenic therapy [244].

MMP-14 is another key molecule in the regulation of angiogenesis. Membrane type 1 (MT1)-MMP belongs to the MT-MMPs’ family that includes transmembrane enzymes specialized in cleaving ECM components adjacent to the cell surface, and its expression by ECs is vital during their migration to form new vessels. However, the action of MT1-MMP is controversial since it can also exert anti-angiogenic effects in a direct or indirect manner. MT1-MMP is highly expressed in CRC [245] and its expression, together with that of α5β1 integrin, which plays a prominent role in angiogenesis [69], positively correlates with CRC progression [246], suggesting a synergistic effect between integrin and this enzyme. MT1-MMP can directly degrade ECM components, but also cleave cell surface molecules, such as extracellular matrix metalloproteinase inducer (EMMPRIN) (CD147, an inducer of MMP expression), low density lipoprotein receptor-related protein (LRP), CD44, and cadherins [247,248,249]. In addition, MT1-MMP is the major activator of MMP-2 and MMP-13 [250] and plays a key role in tube formation during angiogenesis [251]. The angiogenic function of MT1-MMP in the context of the colon microenvironment is scantly documented; nevertheless, some evidence suggests that the response of these degrading enzymes is extremely complex and exert their function not only towards ECM components but also towards other bioactive substrates. For instance, MT1-MMP is known to mediate endoglin shedding [252]. Endoglin plays a crucial role in angiogenesis, is highly expressed on activated ECs, its expression is upregulated in various cancers [253], and is an early indicator of the angiogenic switch in the premalignant lesions of the colon mucosa [254]. Endoglin was also suggested to represent a putative prognostic factor for CRC patients, given the positive correlation with angiolymphatic invasion and metastases to lymph nodes and liver [255]. The shedding of endoglin by MT1-MMP may regulate the angiogenic potential of ECs in the CRC microenvironment [252]. The local upregulation of endothelial MT1-MMP expression increased shed endoglin (sEndoglin) levels, decreased membrane-localized endoglin, and induced EC quiescence. This was in line with the results conducted on 119 CRC patients where lower levels of circulating sEndoglin associated with a higher angiogenic activity [252]. More recently, Lee and co-authors, using a rat colonic cell line as a model, demonstrated that MT1-MMP was responsible for the degradation of the proteoglycan syndecan-2 [256]. Through its N-terminal domain, syndecan-2 selectively promotes a VEGFA-dependent neovascularization enhancing 6-O heparan sulfate chains’ sulfation [257]. Thus, its cleavage by MT1-MMP may significantly impact on angiogenesis.

In the context of GC, MT1-MMP represents a well-established prognostic factor [258,259] and correlates with the invasion capabilities and the metastatic potential of GC cells. However, a direct function of MT1-MMP in GC-associated angiogenesis is not yet conclusive.

Overall, these studies indicate that the role of MMPs in the context of the neovasculature associated with the GI tumor microenvironment is complex and occasionally controversial, and is the result of the balance between pro- and anti-angiogenic factors generated. Interestingly, the use of synthetic inhibitors of proteases, through the blockage of MMPs and of angiogenesis, was proven useful in reducing the metastasis of human colon cancer cells [260]. Besides MMPs, many other proteases are involved in this process, such as plasmin, urokinase, heparanases, and phosphatidyl-inositol phospholipase [261]. These proteases can cleave larger VEGFA isoforms into smaller fragments or, alternatively, mediate VEGFA release from ECM.

Another important class of ECM-degrading enzymes is represented by the ‘a disintegrin and metalloproteinase with thrombospondin motif’ (ADAMTS) family, which takes part in multiple physiological and pathological processes [262]. In the last two decades, the involvement of ADAMTS’ proteins in vascular homeostasis as anti-angiogenic players has been widely documented. Vázquez and colleagues functionally characterized ADAMTS1 and ADAMTS8, demonstrating for the first time that these enzymes were able to specifically inhibit growth factor-mediated neovascularization [263]. ADAMTS1 directly binds to VEGFA through the carboxy-terminal region, suggesting that the anti-angiogenetic activity of ADAMTS1 could be exerted through the modulation of VEGFA bioavailability [264]. In a model of metastatic CRC, an interplay between EC-secreted ADAMTS1 and TSP-1 was identified; ADAMTS1 cleaved TSP-1, releasing anti-angiogenic fragments, which, in turn, act locally to inhibit angiogenesis [265]. Processing of TSP-1 into anti-angiogenic fragments by ADAMTS1 occurs much more efficiently in the liver than in lungs and thus the over-expression of TSP-1 in the liver suppresses the vascularization of liver metastases [265]. The vascular suppression very likely involves an apoptotic mechanism, prompted following the engagement of the EC receptor CD36 by TSP-1 [178].

Overall, ADAMTS1 was reported to impair colon cancer progression and its expression associates with the cancer cell aggressiveness [266]. Accordingly, the expression of the *ADAMTS1* gene is downregulated at any colon cancer stage [266].

The anti-tumor and anti-angiogenic effect displayed by ADAMTS1 has been described also in the context of GC, where a negative correlation between ADAMTS1 and VEGFA mRNA and protein expression was detected [267]. ADAMTS1 protein expression also negatively correlates with the vascular density of primary gastric tumors. In contrast, in the normal gastric mucosa, in primary gastric tumors, and in metastatic lymph nodes, no correlation was detected between ADAMTS1 and TSP-1 mRNA and protein expression, suggesting that the interplay between ADAMTS1 and TSP-1 described in the context of CRC is not prominent in the gastric microenvironment. This evidence further highlights the key role of the microenvironment in determining the different angiogenic properties in CRC and GC, and, thus, the different response to anti-angiogenic therapies.

VEGFA bioavailability is also regulated via the thrombospondin type 1 repeat (TSR1) of ADAMTS5, leading to impaired angiogenesis and tumorigenesis [268,269]. More recently, ADAMTS5 was proposed as an independent prognostic factor for GC since its expression is downregulated by promoter methylation with a consequent increase of GC cell migration and invasive properties, and patients displaying higher ADAMTS5 levels are characterized by a better five-year overall survival rate [270].

Similar results to those observed for ADAMTS5 were reported for other members of the family, such as ADAMTS8 and ADAMTS9. In GC, the methylation status of ADAMTS8 inversely correlates with the protein expression and lower ADAMTS8 levels associate with a higher invasive depth and with the presence of lymph node metastasis [271]. The promoters of ADAMTS8 and ADAMTS9 are methylated also in CRC and this correlates with a decreased expression of these proteases [272,273,274]. Also, the hypermethylation of the *ADAMTS12* gene in CRC associates with the downregulation of its expression and impaired angiogenesis [275]. Thus, in GI cancers, ADAMTS proteins are generally downregulated both at the mRNA and protein levels by promoter methylation, and this correlates with a worse prognosis for the patients. Despite the documented activity of these proteases in the regulation of angiogenesis, in the context of GI tumors it is widely accepted. Only in some studies has an inverse correlation between ADAMTS expression and tumor microvessel density or VEGFA expression been clearly described [267,270,271]. However, the precise molecular mechanism behind the role of ADAMTS proteins in vasculogenesis is still under investigation. It is possible that the anti-angiogenic function of ADAMTS could principally rely on the regulation of cytokines’ bioavailability rather than on the digestive activity. For instance, ADAMTS9 neither cleaved TSP-1 nor TSP-2, nor bound VEGFA [276]. Thus, its endogenous anti-angiogenic activity on ECs is exerted through different molecular mechanisms than those used by the related ADAMTS1 [264,265].

Several studies show that both CRC and GC are highly infiltrated by mast cells [277,278,279,280,281,282] and the presence of these immune cells is predictive of a worse patient outcome. Mast cells are important regulators of the angiogenic process and may be in part responsible for the resistance to anti-angiogenic therapy [283]. Anti-angiogenic therapy stimulates mast cells to produce granzyme b (GZMB), which in turn mobilizes the laminin- and vitronectin-bound FGF-1 and GM-CSF overcoming the blockage of the VEGFA/VEGFR2 signaling axis [283]. Thus, the mast cells adjacent to the vessels through the secretion of GZMB can rapidly supply the growing vasculature with ECM-derived growth factors distinct from the one/s being blocked by the therapy, without the need of time- and energy-consuming protein biosynthesis. However, VEGFA itself can be freed by the ECM remodeling action of GZMB. In fact, in a model of inflammatory disease Hendel and colleagues showed that GZMB induces the release of VEGFA from both fibronectin and EC-derived matrix resulting in a significant increase of vascular permeability [284]. Thus, it is likely that the association between mast cell infiltration and angiogenesis observed in CRC and GC [277,278,279,281,285] could hinge on the increased secretion of GZMB and the consequent release of pro-angiogenic factors.

A distinct mechanism, other than ECM degradation, by which ECM remodeling affects angiogenesis is mediated by the accumulation of ECM molecules generated by the activity of tissue transglutaminase-2 (TG2), a multifunctional enzyme that catalyzes the multimerization of proteins by generating e(g-glutamyl)lysine isopeptide bonds [286]. The cross-linking action of TG2 is exerted towards many ECM proteins and the activity of this enzyme in ECs is thought to be important for the stabilization of the basement membrane [287,288]. TG2 is downregulated in ECs undergoing capillary morphogenesis, which requires localized destabilization of the matrix [289], suggesting that this molecule is an important regulator of ECM deposition and stabilization. Jones and colleagues demonstrated that the injection of TG2 in colon carcinoma tumors resulted in overall delay of tumor growth and complete regression of 20% of the tumors [290]. The decreased tumor growth associated with impaired vascularization, due to the accumulation of ECM molecules and increased cross-linking, compromises EC tube formation and reduces angiogenesis in a dose-dependent manner. As a result, it alters the homeostasis of ECM turnover towards deposition rather than the initial destabilization of the matrix, which is a prerequisite for EC proliferation and migration. The mechanisms involved in ECM remodeling in GI tumor-associated angiogenesis are schematically represented in Figure 3.

The literature pinpoints the ECM remodeling as an important actor in the vascularization of the GI tumor microenvironment, but an extensive analysis is still required to better understand the mechanisms of action that may grant the possibility to develop new therapies to improve the outcome of GI cancer patients.

## 5. ECM and Growth Factors in a Tangled Crosstalk

As previously mentioned, the ECM is a key regulator of growth factor availability [291], and a deep understanding of the crosstalk between ECM and cytokines could entail new opportunities to anti-angiogenic therapy in GI tumors.

VEGFA is the best known pro-angiogenic factor and plays a crucial role during tumor vessels’ formation [292]. Thus, many therapeutic strategies aimed at blocking this cytokine to starve tumors have been proposed. Metastatic CRC patients can benefit from the VEGFA humanized blocking antibody bevacizumab [34], and, in second line, from the chimeric VEGFR1/VEGFR2-based decoy receptor VEGFA-Trap aflibercept [293]. Furthermore, patients with advanced GI stromal tumors in second-line therapy can take advantage of sunitinib, a VEGFR2 inhibitor [294]. More recently, the VEGFR2-blocking antibody ramucirumab has been introduced in the treatment of metastatic CRC [295] and metastatic GC [296]. However, many patients are refractory or insensitive to anti-angiogenic therapy [297] and a better understanding of the TME may grant the possibility to overcome this problem. Rahbari and colleagues demonstrated that VEGFA-targeted therapy in metastatic CRC exacerbates the hypoxic TME, leading to hyaluronic acid and sulphated glycosaminoglycans (sGAGs) accumulation. As a consequence, increased ECM stiffness may compromise vascular perfusion, drug delivery, and, thus, therapy efficacy [298,299]. This elegant study highlighted the need to better understand the complex crosstalk between VEGFA and ECM in order to improve anti-angiogenic therapy in GI tumors.

As dealt with in the dedicated sections, proteases such as MMPs and ADAMTS, but also heparanases and plasmin, are important contributors during angiogenesis, also thanks to the mediation of VEGFA release from the ECM and the production of active VEGFA fragments [300]. The heparan sulphate proteoglycan perlecan sequesters VEGFA, regulating its availability along the vascular BM, thus controlling VEGFR2 activation [79]. Consistently, perlecan expression is down-modulated in colon cancer [301] and in GC its reduction could contribute to ECs’ resistance to anoikis during therapy [302]. Instead, as discussed in the dedicated section, multimerin-2 sequesters VEGFA, affecting VEGFR2 angiogenic response [116,117,118]. Indeed, tumor-bearing mice over-expressing multimerin-2 showed impaired vascularization and reduced tumor growth [116,117], and, on the other hand, multimerin-2 expression was lower in tumor-associated vessels in GC patients with respect to the normal mucosa counterpart [119]. On the contrary, other ECM components are upregulated in TME. For instance, the increased expression of the small leucine-rich repeat proteoglycan byglican associates with poor prognosis in GC patients. Byglican induces VEGFA expression in ECs, boosting angiogenesis and GC growth [303]. An indirect increase of VEGFA occurs via the recruitment of cancer associated fibroblasts (CAFs) prompted by the upregulation of laminin α1, which can also bind the cytokine, further enhancing the pro-angiogenic response [42]. The expression of VEGFA, as well as FGF-2, is also triggered by the fibronectin extra-domain B, the tumor-specific isoform of fibronectin, and was shown to increase esophageal cancer vascularization [304]. FGF-2 is a potent inducer of angiogenesis [305,306], can be sequestered by both TSP-1 and TSP-2, and hinders the activation of its receptor, suggesting a possible strategy to overcome resistance to anti-angiogenic therapy [307,308,309]. Furthermore, Fuster and colleagues demonstrated that interfering with the interaction between heparan sulfate and VEGFA/FGF-2 may potentially represent a good strategy to target tumor-associated ECs, at least in lung cancer model [310]. However, this approach could be applied also in GI tumors, given that the expression of heparan sulphate proteoglycans is often altered in these tumors. Of note, FGF-2 upregulation is considered one of the possible mechanisms of acquired resistance to canonical anti-VEGFA therapy [311,312]. The FGF-2 levels rise with disease progression and resistance to bevacizumab [313], suggesting that the blockage of both cytokines may be beneficial. In line with this hypothesis, the use of regorafenib, a tyrosine kinase receptor inhibitor able to block important angiogenic receptors, including FGFR and VEGFR2, is beneficial in the treatment of metastatic CRC and GC [314,315].

Platelet-derived growth factor B (PDGF-B) is a prominent regulator of angiogenesis, acting autocrinally on ECs and mediating pericytes’ recruitment [316]. PDGF-B can be retained adjacent to the EC BM proteins, such as collagens, laminins, perlecan, and nidogen, favoring pericyte migration and association to the vessel wall. However, despite tumors often display high PDGF levels, cancer-associated vessels exhibit poor pericyte coverage, possibly due to the increased availability of pro-angiogenic factors, which interfere with a proper BM deposition and, thus, PDGF localization. In addition, cancer cells frequently express platelet-derived growth factor receptor (PDGFR), becoming sensitive to the PDGF proliferative and migratory stimuli, and in GI cancers the upregulation of PDGF and PDGFR associates with a poor outcome [317,318]. Hosaka and colleagues demonstrated that increased levels of tumor cell-derived PDGF induces the internalization of PDGFR in pericytes, which then fail to properly adhere to BM proteins [319]. In fact, a combined treatment with bevacizumab and the RTK inhibitor, imatinib, which among other receptors blocks PDGFR, induces proper deposition of collagen IV and vessel normalization in colon cancer [320].

Angiopoietins (ANGs) regulate EC homeostasis engaging the Tie-2 receptor, and the most studied members, ANG-1 and ANG-2, display antagonizing effects: ANG-1 stabilizes and ANG-2 supports vessel remodeling, permeability, and proliferation [321]. ANG-1 is mainly stored in the ECM adjacent to the EC surface, whereas ANG-2 acts more as a soluble factor [322]. ANG-2 is frequently upregulated in CRC and GC and associates with increased metastatic potential [323,324]. Accordingly, the blockage of ANG-2 reduces CRC growth, likely due to vessel normalization [325], and may improve anti-VEGFA therapy in combinatorial settings [326]. The use of vanucizumab, a bispecific antibody able to block both VEGFA and ANG-2, has been tested in a phase I clinical trial in a cohort of advanced CRC patients with encouraging results [327]. The main therapies targeting angiogenic growth factors are reported in Table 1.

Overall, these studies demonstrate that a better understanding of the crosstalk between the ECM and angiogenic active growth factors may guide the development of new, promising, therapeutic approaches for GI and other cancers.

## 6. Conclusions

It is widely accepted that the growth and progression of tumors does not depend only on the nature and mutation status of tumor cells. These cells are part of a complex microenvironmental niche, which does not represent a mere bystander during tumor development. Instead, it can profoundly affect progression and metastatic spread, as well as therapy efficacy. In particular, the vascularization of the tumors and the efficiency and quality of intratumoral vessels play a key role, not only in affecting tumor growth but also in influencing hematogenous metastatic spread and drug delivery/efficacy. As summarized here, the ECM is a master regulator of angiogenesis and vascular stability, and its key role has also been highlighted in the context of GI cancers. Many are the molecules involved in these phenomena, and a further layer of complexity is due to the fact that the ECM is not a stable component; on the contrary, it is continuously rearranged, which leads to the formation of protein fragments/peptides that may entail angiogenic properties often antagonizing those of the molecule of origin. The balance of these components and their expression levels within the TME will finally determine the tumor fate. Since, in GI tumors, the matrix composition changes tremendously during development, many questions remain unanswered and intensive research will be necessary to try develop new multi-target therapies aimed at normalizing the ECM composition to impair the metastatic spreading and improve drug delivery and the response of GI cancer patients.

## Figures and Tables

**Figure 1 ijms-21-03686-f001:**
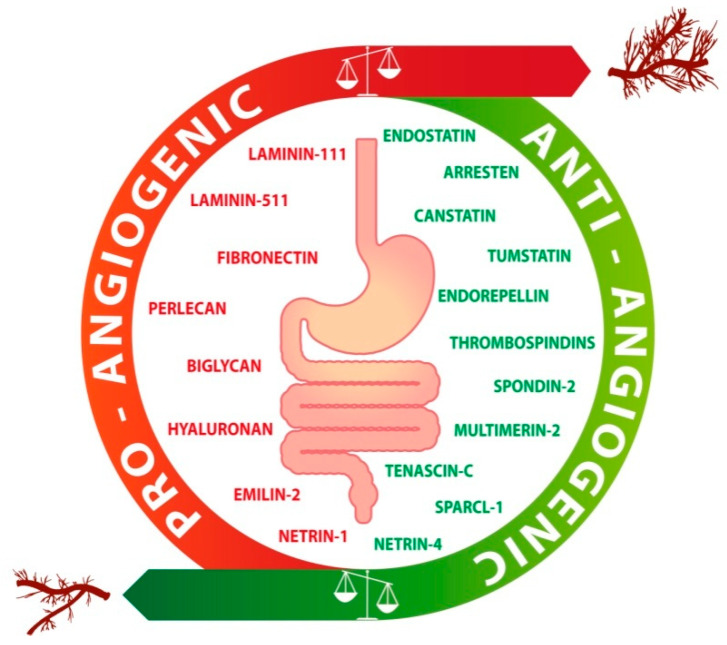
Schematic drawing of the angiogenic switch in gastrointestinal cancers. The angiogenic switch occurs upon an imbalance between pro-angiogenic (red) and anti-angiogenic (green) molecules. The scheme reports the major extracellular matrix molecules regulating angiogenesis in the context of GI tumors.

**Figure 2 ijms-21-03686-f002:**
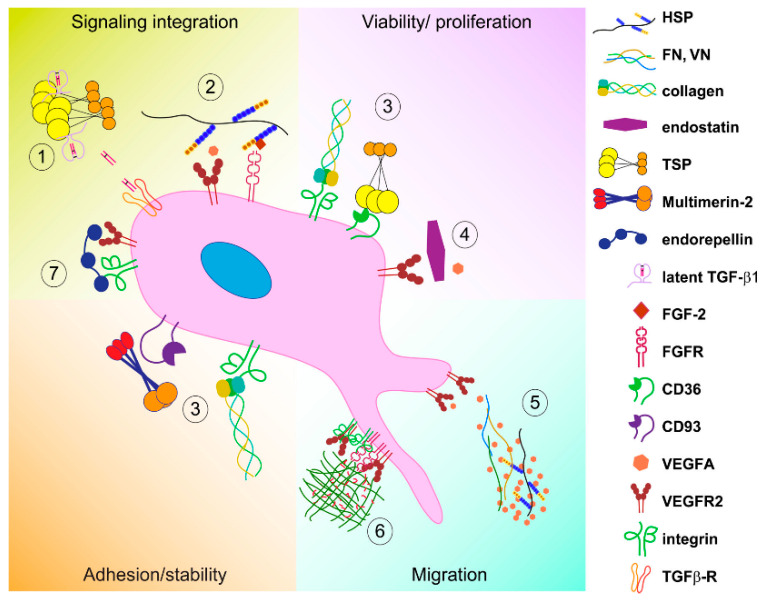
Schematic representation of the major mechanisms by which ECM affects endothelial cell function. The ECM impacts on EC function via these major mechanisms: (**1**) Acting as a modulator of growth factors’ maturation, (**2**) displaying multiple binding sites for different growth factors within the same molecule, thus contributing to receptor clustering and signaling network, (**3**) affecting EC viability and proliferation, engaging cell surface receptors, (**4**) acting as stumbling block for the ligand/receptor interaction, (**5**) regulating the spatio-temporal growth factors’ availability, (**6**) serving as mechanotransducers, and (**7**) simultaneous binding of different receptors and modulating of their trafficking.

**Figure 3 ijms-21-03686-f003:**
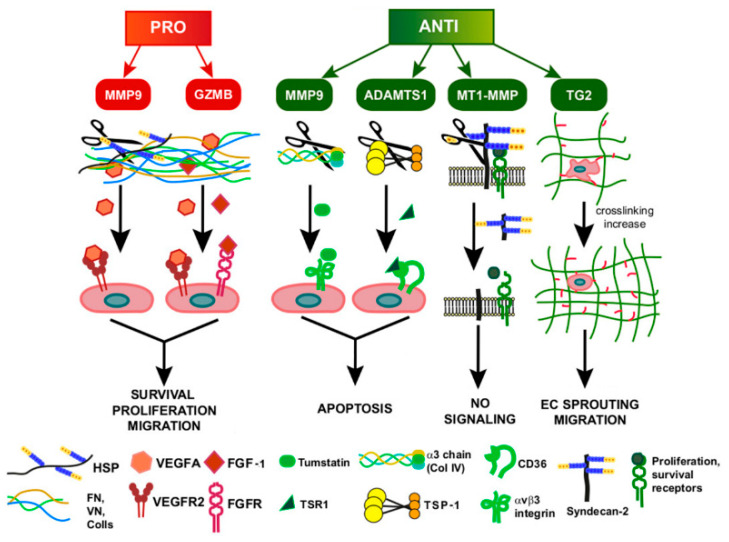
Schematic representation of the role of ECM remodeling in GI tumor-associated angiogenesis. The drawing exemplifies the action of proteases in affecting EC behavior and leading to pro-angiogenic (red) or anti-angiogenic (green) effects. The scheme summarizes the major molecules exerting these functions in the context of GI tumors.

**Table 1 ijms-21-03686-t001:** Summary of the major drugs targeting ECs, currently used for the treatment of GI tumors.

Antiangiogenic Drug	Target	Type of Molecule	Cancer Type	Clinical Use	Ref
**Bevacizumab**	VEGFA	Blocking humanized mAb	mCRC	1st line alone or in combination with chemotherapy	[34]
**Aflibercept**	VEGFA	Trapping Recombinant fusion protein (VEGFR1-VEGFR2)	mCRC	2nd line	[293]
**Sunitinib**	VEGFR2	Small RTK inhibitor	GC	2nd line	[294]
**Ramucirumb**	VEGFR2	Blocking humanized mAb	mCRC/advanced GC	2nd line alone or in combination with chemotherapy	[295,296]
**Regorafenib**	FGFR2/VEGFR2	Small RTK inhibitor	mCRC/GC	After the failure of other lines of therapies	[314,315]
**Imatinib**	PDGFR	Small RTK inhibitor	mCRC/advanced GC	Adjuvant chemotherapy in patients with *wt* PDGFR	[320]
**Vanucizumab**	ANG2/VEGFA	Bi-specific mAb	CRC	Not approved	[327]

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
