# Peer review of "Role of Extracellular Matrix in Gastrointestinal Cancer-Associated Angiogenesis"

_ijms, 2020, doi:10.3390/ijms21103686_

Round 1
Reviewer 1 Report
I read with interest the review work of Andreuzzi et al. The authors in this extensive work described the complex nature of the extracellular matrix focusing on its role in gastrointestinal cancer, and in particular its role in the process of angiogenesis. The authors reviewed a wide collection of literature on this topic, citing "historical" and recent publications. Because the work is extensive, it is good that the authors made illustrations summarizing some chapters, as well as a table summarizing the anti-angiogenic therapies used in GI.
I have some minor comments:
The authors should enlarge Fig. 1 because the names of the ECM molecules are illegible.
A very extensive chapter 2 on ECM molecules could benefit if the described families of ECM components and their representatives were collected in tabular form. This would be a good summary of this chapter and could highlight the conclusion drawn by the authors at the end of chapter 2.
The text requires minor editorial corrections, e.g. strikethrough text in lines 31, 669, 692, 711.
Author Response
Response to Reviewer 1 Comments
We thank Reviewer 1 for appreciating the manuscript. In the revised version we have addressed all the minor concerns raised by the Reviewer:
POINT 1) Reviewer: The authors should enlarge Fig. 1 because the names of the ECM molecules are illegible.
As suggested by the Reviewer we have increased the font of the ECM molecules and are now more visible.
POINT 2) Reviewer: A very extensive chapter 2 on ECM molecules could benefit if the described families of ECM components and their representatives were collected in tabular form. This would be a good summary of this chapter and could highlight the conclusion drawn by the authors at the end of chapter 2.
As also suggested by Reviewer 2 this chapter was lengthy, thus we have shortened it and believe that in the revised version is more fluid. We also highlighted each molecule in bold to guide the reader should he/she be more interested in a specific molecule. With these modifications we consider that the introduction of an additional table could be avoided, also in the light of the fact that all the molecules and the relative functions are reported in Fig. 1.
POINT 3) Reviewer: The text requires minor editorial corrections, e.g. strikethrough text in lines 31, 669, 692, 711.
We have introduced the editorial corrections as suggested by the Reviewer.
Reviewer 2 Report
The review Role of Extracellular Matrix in Gastrointestinal Cancer-associated Angiogenesis summarizes the current knowledge on the role of the ECM in CRC. The ECM acts directly on endothelial cells, and indirectly through its remodeling and releases growth factors as a consequence of it. The author reviewed the role of ECM and of its remodeling during cancer progression as an importance for the development of new more efficacious targeted therapies.
Although it is a review, 363 references are way too much. Some of them are out dated and could get replaced through a recent review or publications.
For example in row 53-56 the author uses 11 references for this sentence
An increasing number of studies have demonstrated that the nature and composition of the tumor microenvironment (TME) can profoundly affect tumor progression, and represents an important mean for the development of new therapeutic approaches to overcome this clinical stumble [9-20].
Otherwise the review is clear written and shows the current research for the role on the extracellular matrix in colorectal and gastric cancer. However some of the paragraphs are too long and it is getting very difficult to read through those paragraphs. There is too much information in it for example 2. ECM and angiogenesis could be shortened.
In this stage it is a little bit difficult going through this review, although there is a lot of good information in this review.
Author Response
Response to Reviewer 2 Comments
We thank Reviewer 2 for the kind comments about the manuscript. As detailed below we have addressed all the minor concerns raised.
POINT 1) Although it is a review, 363 references are way too much. Some of them are out dated and could get replaced through a recent review or publications. For example in row 53-56 the author uses 11 references for this sentence. An increasing number of studies have demonstrated that the nature and composition of the tumor microenvironment (TME) can profoundly affect tumor progression, and represents an important mean for the development of new therapeutic approaches to overcome this clinical stumble [9-20].
As suggested by the Reviewer we removed several references (33) and gave priority to more recent studies.
POINT 2) Otherwise the review is clear written and shows the current research for the role on the extracellular matrix in colorectal and gastric cancer. However some of the paragraphs are too long and it is getting very difficult to read through those paragraphs. There is too much information in it for example 2. ECM and angiogenesis could be shortened.
Indeed this chapter was very lengthy. We have shortened it as suggested by the Reviewer and believe that thanks to this suggestion the chapter is more flowing in the revised version.